# DRL: Discriminative Representation Learning for Class Incremental Learning

## Abstract

Non-rehearsal class incremental learning (CIL) is pivotal in real-world scenarios such as data streaming applications and data security. Despite the remarkable progress in research on CIL, it remains an extremely challenging task due to three conundrums: increasingly large model complexity, non-smooth representation shift during incremental learning and inconsistency between stage-wise sub-problem optimization and global inference. In this work, we propose the Discriminative Representation Learning (*DRL*) method to deal with these challenges specifically. To conduct incremental learning effectively and yet efficiently, our *DRL* is built upon a pre-trained large model with excellent representation learning capability, and increasingly augments the model by learning a lightweight adapter with a small amount of parameter learning overhead in each incremental learning stage. While the adapter is responsible for adapting the model to new classes of data involved in current learning stage, it can inherit and propagate the representation capability from the current model via parallel connection between them. As a result, such design can guarantee a smooth representation shift between different stages of incremental learning. Furthermore, to alleviate the issue of the training-inference inconsistency induced by the stage-wise sub-optimization, we design the Margin-CE loss, which imposes a hard margin between classification boundaries to push for more discriminative representation learning, thereby narrowing down the gap between stage-wise local optimization over a subset of data and global inference on all classes of data. Extensive experiments on six benchmarks reveal that our *DRL* consistently outperforms other state-of-the-art methods throughout the entire CIL period while maintaining high efficiency in both training and inference phases.

## 1 Introduction

Deep neural networks have achieved great improvement in many fields He et al. (2015); Ren et al. (2016); Ran et al. (2022); Zhan et al. (2022); Li et al. (2022), and the characteristic training process of deep neural networks is supervised or self-supervised learning He et al. (2021) with pre-collected datasets (e.g., ImageNet Deng et al. (2009)). However, this conventional process struggles with scenarios where the training data is in a streaming format Dong et al. (2022); Ning et al. (2021) , necessitating incremental learning Zhou et al. (2023a; 2024b), such as class incremental learning (CIL) Tian et al. (2023); Zhao et al. (2023); Li et al. (2023), task incremental learning Van de Ven et al. (2022); Abati et al. (2020), incremental object detection Zhang et al. (2024b), etc. Among the aforementioned methods, non-rehearsal CIL Li & Hoiem (2017); Rebuffi et al. (2017); Zhu et al. (2021) becomes critical, especially in the sequence or privacy-sensitive scene Dong et al. (2022); Shokri & Shmatikov (2015); Chamikara et al. (2018). The objective of non-rehearsal CIL is to acquire new knowledge yet not forget the old one. However, it suffers lower discriminal representation and poor performance, as it cannot access previous datasets, leading to catastrophic forgetting French (1999).

To mitigate the catastrophic forgetting in non-rehearsal CIL, numerous methods have been proposed. Kirkpatrick et al. introduced regularization-based methods that incorporate explicit regularization terms to balance the old and new knowledge by keeping the unified model parameters close to the learned ones, such as EWC Kirkpatrick et al. (2017) and some more advanced versions Ritter et al. (2018); Schwarz et al. (2018); Chaudhry et al. (2018a).

However, these methods do not effectively inherit the capabilities of the previous model, resulting in an inability to alleviate the problem of catastrophic forgetting. Additionally, the added constraints may reduce the model's plasticity Yan et al. (2021) too. Some researchers utilize dynamic network architecture according to the training stage to balance the catastrophic forgetting and plasticity, such as combining multiple networks Aljundi et al. (2017), iterative pruning Mallya & Lazebnik (2018), dynamically expanding sub-network Yoon et al. (2017); Schwarz et al. (2018); Douillard et al. (2022), etc. Among these methods, DER Yan et al. (2021) preserves the previously trained model to alleviate catastrophic forgetting and expands a new model for each stage. FOSTER Wang et al. (2022a), recognizing the excessive number of models in inference for DER, employs knowledge dis-

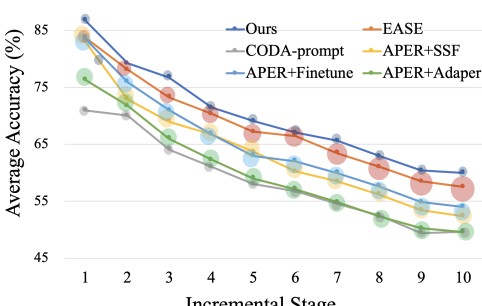

Figure 1: Performance comparison in terms of both classification accuracy and inference complexity by model size between different methods on ImageNet-A B0 Inc20. The size of circles denotes the model size during inference.

tillation (KD) to compress the model and limit its size. However, this approach requires additional model parameters and a complex training process. Relying solely on KD to inherit the capabilities of the previous model has a limited effect on reducing forgetting. Recently, many researchers Zhou et al. (2024b); Zheng et al. (2023); Wang et al. (2022d) have insight that the use of large Pre-Trained Model (PTM) can significantly improve the CIL performance. Building on PTM, Zhou et al. proposed EASE Zhou et al. (2024c), which retains all trained models in memory to alleviate catastrophic forgetting and expands an independent PTM with a learning adapter Hu et al. (2022) to acquire new knowledge. However, the models stored in memory are cumbersome (see Figure 1) and suboptimal due to the lack of interaction between different stages during training. Additionally, current methods widely use cross-entropy loss (CELoss) for supervision during each training stage. A potential problem with this training approach is the inconsistent separation granularity between the training and inference phases, which has yet to be resolved.

To address the aforementioned problems, we propose a discriminative representation learning method consisting of an Incremental Parallel Adapter (*IPA*) network and a Margin Cross-Entropy Loss (Margin-CE loss), which achieves a better stability-plasticity trade-off with high efficiency. Our *IPA* is built upon a PTM and dynamically expands a parallel adapter for each stage. Furthermore, we find that the features of the trained model are beneficial for the current stage. For instance, essential features representing a 'dog' can also assist in defining a 'cat.' Therefore, we propose a learning transfer gate to selectively inherit this robust representation ability. Thanks to this gate, we can achieve strong plasticity with exceptional efficiency. To alleviate the issue of training-inference inconsistency induced by stage-wise sub-optimization, Margin-CE loss imposes a margin between the classification boundaries for different classes to optimize inter-class separability, thereby yielding more discriminative representation learning.

Finally, we carried out experiments on six benchmark datasets, and the results verified the state-of-the-art (SOTA) performance of *DRL*. On ImageNet-A, our method achieves an accuracy of 68.79%, which is 3.45% higher than the current SOTA. On VTAB, ObjectNet, we achieve 95.73%, 72.69% accuracy, and 2.12%, 1.85% higher than the current SOTA. Our main contributions are:

- We propose a novel incremental parallel adapter network that achieves a better stability-plasticity trade-off with high training and inference efficiency. The *IPA* is established on PTM. It achieves superior plasticity through an parallel adapter and a learnable transfer gate, while mitigating catastrophic forgetting by isolating the trained parameters.

- Furthermore, we propose a margin cross-entropy loss to enhance the discriminative representation ability by mitigating the inconsistency between stage-wise sub-problem optimization and global inference. The Margin-CE loss is simple, yet effective, and can be seamlessly integrated into other methods.

- Our approach achieves new state-of-the-art performance on all six benchmarks, including commonly CIL benchmarks and out-of-distribution benchmarks which have large domain gaps from pre-trained model's datasets.

The rest of this paper is organized as follows. First, we investigate the current CIL methods in Section 2. Followed by the presentation of the preliminaries in Section 3.1, and the introduction of our *IPA* and Margin-CE loss in Sections 3.2 and 3.3 respectively. Comprehensive evaluations are exhibited in Section 4. Finally, We conclude the paper with a summary of our method.

## 2 RELATED WORKS

**Class incremental learning.** CIL is essential in data streaming application scenarios, where the learning system is required to continually incorporate new class knowledge without forgetting existing ones Zhou et al. (2023a); Wang et al. (2023c); Zhuang et al. (2023; 2022); Liu et al. (2021); Zhao et al. (2021a); Dong et al. (2022); Gao et al. (2022); Wang et al. (2023a); Goswami et al. (2023). Based on the accessibility of a portion of the training data from previous stages, these methods are categorized into rehearsal-based Aljundi et al. (2019b); Liu et al. (2020); Zhao et al. (2021b); Chaudhry et al. (2018b) and non-rehearsal CIL Douillard et al. (2020); Simon et al. (2021); Tao et al. (2020); Kirkpatrick et al. (2017); Aljundi et al. (2019a; 2018); Zenke et al. (2017); Zhao et al. (2020); Yu et al. (2020); Shi et al. (2022); Pham et al. (2022). Rehearsal-based methods address catastrophic forgetting by retaining a small set of old training examples in memory. However, storing exemplars of old tasks is not always desirable due to data security and privacy concerns Shokri & Shmatikov (2015); Ning et al. (2021); Dong et al. (2022). Consequently, many researchers have shifted their focus to non-rehearsal CIL, which fine-tunes the model without relying on exemplars. Some of these researchers have proposed regularization-based methods, such as EWC Kirkpatrick et al. (2017) and some more advanced versions Ritter et al. (2018); Schwarz et al. (2018); Chaudhry et al. (2018a). These methods introduce explicit regularization terms to balance old and new knowledge by constraining the unified model parameters to remain close to the learned values. However, these methods do not effectively inherit the capabilities of the previous model, resulting in an inability to alleviate the problem of catastrophic forgetting. Additionally, the added constraints may reduce the model's plasticity too.

**Dynamic network-based methods.** As representatives of non-rehearsal learning methods Qu et al. (2021), dynamic network-based methods address catastrophic forgetting by allocating specific model parameters to each stage. Recently, expandable networks Yan et al. (2021); Wang et al. (2022a); Douillard et al. (2022); Chen & Chang (2023); Hu et al. (2023); Huang et al. (2023) have demonstrated strong performance among their competitors. However, many of these methods rely on expanding the backbone network or large modules, resulting in cumbersome networks after multiple incremental stages, which lack flexibility and efficiency. In contrast, our method utilizes small parallel sub-networks and adopts different strategies to enhance network efficiency, and significantly improve the performance with little computational cost.

**Pre-trained model-based methods.** Pre-trained model-based (PTM-based) methods, which particularly leverage the PTM's strong representational capabilities, have become a hot topic recently Zhou et al. (2024b); Wang et al. (2023b); McDonnell et al. (2024). These methods are generally divided into prompt-based Wang et al. (2022c;d); Smith et al. (2023); Wang et al. (2022b) and adapter-based approaches. Recently, L2P Wang et al. (2022d) and DualPrompt Wang et al. (2022c) have utilized prompt tuning based on PTM for incremental learning tasks. However, these methods still utilize a unified prompts pool which needs to be updated. This action directs to prompt-level forgetting and representation ability will be restricted. Other methods, such as APER Zhou et al. (2024a) and EASE Zhou et al. (2024c), expand an independent PTM with a trainable adapter Houlsby et al. (2019); Hu et al. (2022) to fine-tune the model for each stage and employ a prototype-based classifier to maintain the generalizability of PTM in inference. However, these methods utilize all stage models in inference, which is inefficient and inadequate due to the lack of interaction between models at different stages.

**Loss functions in CIL.** Existing CIL methods employ various loss functions for supervision. Typical methods involve classification losses for recognition (e.g., cross-entropy loss Zhou et al. (2024a;c); Smith et al. (2023); Wang et al. (2022d)), regularization losses (e.g., distillation loss Wen et al. (2024); Li et al. (2024); Li & Hoiem (2017); Hinton et al. (2015), elastic weight consolidation loss Kirkpatrick et al. (2017); Magistri et al. (2024), or gradient-based loss Elsayed & Mahmood (2024)), as well as proxy losses for sub-module objectives (e.g., feature selection and discriminative enhancement losses Wang et al. (2022d); Douillard et al. (2022); Zhang et al. (2024a); Goswami

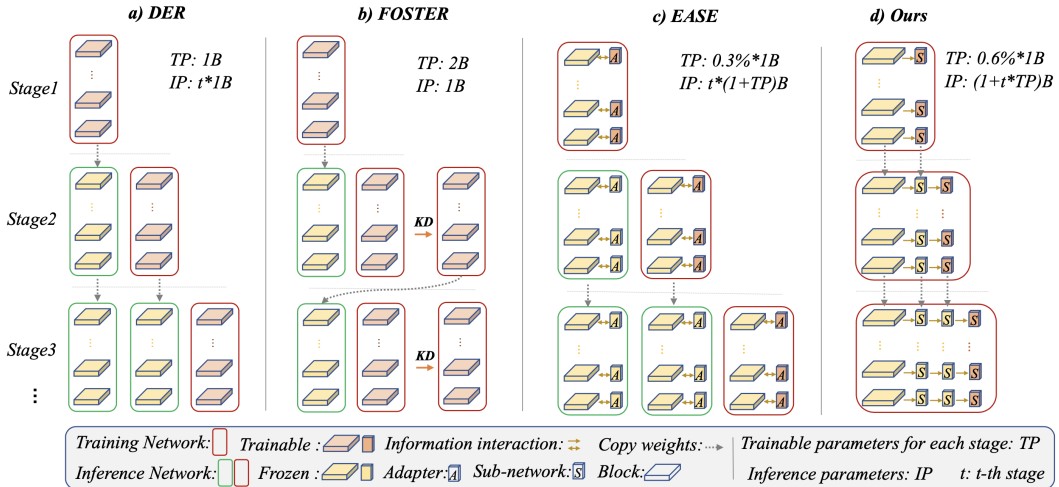

Figure 2: Comparison of different CIL methods. We select the three representative methods, DER, FOSTER, and EASE. '1B' denotes the total parameters of a model (e.g., ViT-B/16). a) DER creates a new model for each stage, while b) FOSTER utilizes KD to compress the model to limit the model size. c) EASE expands a PTM with a learning adapter to reduce the trainable parameters for each stage. However, both DER and EASE require maintaining all the trained models in memory for inference, which is cumbersome. FOSTER necessitates an extra training step, and the compressing process brings further catastrophic forgetting. d) shows our network, which inherits the old feature to eliminate catastrophic forgetting and expands an adapter with few trainable parameters to learn new knowledge. It utilizes the unified model to predict which is more efficient for inference.

et al. (2024)). In the previous discussion, softmax cross-entropy loss (CELoss) is widely used in a stage-wise manner across various methods Zhou et al. (2024a;c). However, a potential problem with this training approach is the inconsistent separation granularity between the training and inference phases. Our proposed Margin-CE loss can eliminate this inconsistency, resulting in significant performance improvement.

In summary, our *DRL* addresses the limitations of existing incremental learning methods by proposing *IPA* and Margin-CE loss to improve the model representation. This approach offers improved performance and resource efficiency compared to existing methods, achieving better stability-plasticity trade-off in the CIL.

## 3 METHOD

### 3.1 PRELIMINARIES

CIL aims to train a model with the training samples arriving in sequence. This incremental process can be divided into $T$ stages. For the stage $t \in \{1,2,...,T\}$, the training samples belonging to the stage $t$ are represented as $D^t = \{X^t, Y^t\}$, where $X^t$ is the input data, and $Y^t$ corresponds to the associated label. The classes across different stages do not overlapped, i.e., $Y^1 \cap Y^2 \cap ... \cap Y^T = \emptyset$. The non-rehearsal CIL satisfies $D^1 \cap D^2 \cap ... \cap D^T = \emptyset$. During the training of the model at the $t$-th stage, We can only access the data $D^t$, while the stage identity $t$ is not available during inference. After each stage, the trained model is evaluated on all previously seen classes, i.e., $Y^1 \cup Y^2 \cup ... \cup Y^t$.

The modeling of CIL can be formulated as $f(\mathbf{x}) = X \rightarrow Y$, which aims to minimize the empirical risk:

$$\sum_{(\mathbf{x},y) \in D^1...\cup D^T} L(f(\mathbf{x}), y) \tag{1}$$

Here, we decouple our model into the embedding module $\Phi(\cdot) : \mathbb{R}^D \rightarrow \mathbb{R}^d$ and classifier layer $\mathbf{W} \in \mathbb{R}^{d \times |Y|}$, where $d$ represents the embedding dimension and $Y$ represents the label space. The model output is then denoted as $f(\mathbf{x}) = \mathbf{W}^\top \Phi(\mathbf{x})$. Since our *DRL* is based on PTM, for the $t$-th stage, the embedding module can be further parameterized $\Theta = \{\theta_t^o, \theta_t^n\}$, where $\theta_t^o$ and $\theta_t^n$ are the

parameters of the trained model (e.g., PTM) and the new expanding network (e.g., adapter Zhou et al. (2024c)) respectively. Furthermore, for the $l$-th transformer block Dosovitskiy (2020), where $l \in \{1, ..., L\}$ and $L$ represents the total number of blocks (e.g, $L = 12$ in ViT-B/16), the parameters are denoted as $\{\theta_t^{o_l}, \theta_t^{n_l}\}$. The classifier layer can be further decomposed into a combination of $\mathbf{W} = [\mathbf{w}_1, ..., \mathbf{w}_{|Y|}]$. The classifier weight for class $i$ is $\mathbf{w}_i$ and , $\mathbf{w}_i \in \mathbb{R}^{d \times 1}$.

Following the EASE Zhou et al. (2024c), in the training phase, the logit for the class $i$ is:

$$z_i = s \cdot \cos(\mathbf{w}_i, \Phi(\mathbf{x})) \tag{2}$$

Where $s$ is a learnable scale factor during the training phase. The logit $z_i$ is passed to the softmax function to obtain the output probability:

$$p_i = \frac{e^{z_i}}{\sum_j e^{z_j}} = \frac{e^{s \cdot \cos(\mathbf{w}_i, \Phi(\mathbf{x}))}}{\sum_j e^{s \cdot \cos(\mathbf{w}_j, \Phi(\mathbf{x}))}} \tag{3}$$

During inference, the prototype-based classifier extracts the final [CLS] token as the class center $\mathbf{c}_i$ (i.e., prototype) for the $i$-th class and directly replaces the $\mathbf{w}_i$, it then utilizes cosine distance to calculate the predicted probability, as follows:

$$\hat{p}_i = \cos(\mathbf{c}_i, \Phi(\mathbf{x})) = \frac{\mathbf{c}_i^\top \Phi(\mathbf{x})}{\|\mathbf{c}_i\|_2 \cdot \|\Phi(\mathbf{x})\|_2} \tag{4}$$

### 3.2 INCREMENTAL PARALLEL ADAPTER

PTM-based methods demonstrate promising performance in CIL. Consequently, many researchers Wang et al. (2022a;c) have sought to make slight adjustments to PTMs, such as APER and EASE. However, these methods either suffer from a poor stability-plasticity trade-off Wang et al. (2022d;c) or a cumbersome structure during inference Yan et al. (2021); Wang et al. (2022a); Zhou et al. (2024c). Here, we propose Incremental Parallel Adapter (*IPA*) to alleviate those problems. Building on PTM, *IPA* achieves a better stability-plasticity trade-off with high efficiency. The details of *IPA* are shown in Figure 3 which mainly consists of three parts: the trained network parameterized by $\theta^o = \{\theta^{o_1}, ...\theta^{o_l}...\theta^{o_L}\}$, the efficient sub-network parameterized by $\theta^e = \{\theta^{e_1}, ...\theta^{e_l}...\theta^{e_L}\}$, and the learnable transfer gate parameterized by

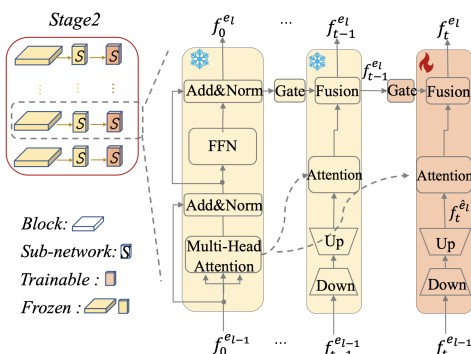

Figure 3: Details of each block in our *IPA*.

$\theta^g = \{\theta^{g_1}, ...\theta^{g_l}...\theta^{g_L}\}$. Consequently, $\theta^n = \{\theta^e, \theta^g\}$. For the $l$-th block during training, where $l \in \{1, ..., L\}$, the $\theta^{o_l}$ is fixed while $\theta^{n_l}$ is learnable. In each training stage of CIL, we freeze the trained model in previous stage, and augment it with a new learnable Incremental Parallel Adapter. In particular, we design a transfer gate to connect two *IPA* adapters between two adjacent stages for smooth representation shift, as shown in Figure 3.

The trained model in previous stage is utilized to extract fundamental features by freezing the trained parameters, which can be either a PTM or a model trained in previous stage. Freezing the parameters $\theta^o$ helps to retain the representation ability of the trained model and effectively reduces catastrophic forgetting. Following the APER and EASE, we utilize the Vision Transformer (ViT), pre-trained on ImageNet Deng et al. (2009), as the trained network(i.e., ViT-B/16-IN21K Dosovitskiy (2020)) in the first stage. Trained with over 11 million images across 21,000 categories, the ViT-B/16-IN21K offers strong representational capabilities and enhances the discriminative power for incremental tasks.

The newly inserted IPA adapter is utilized to learn new knowledge, and each sub-network is dynamically expanded with each new stage. To prevent the network from becoming cumbersome after multiple incremental stages, each block is designed as a small, trainable module. Specifically, the sub-network consists of an adapter and an attention module. The adapter is a 1x1 convolutional layer

$\mathbf{W}_{down} \in \mathbb{R}^{d \times r}$ for downsampling, followed by an activation function, and another 1x1 convolutional layer $\mathbf{W}_{up} \in \mathbb{R}^{r \times d}$ for upsampling. This bottleneck-like structure, consisting solely of two 1x1 convolutions, makes the adapter extremely lightweight. The input to the $l$-th adapter for the $t$-th task is the output from the $l$-1 block (e.g., $\boldsymbol{f}_t^{e_{l-1}}$ in Figure 3), and the output is denoted as $\boldsymbol{f}_t^{\hat{e}_l}$. The attention module aims to enhance the correlations between features (or tokens). Traditionally, the attention module Vaswani et al. (2023) requires three additional 1x1 convolutional layers to generate the $\mathbf{Q}$, $\mathbf{K}$, $\mathbf{V}$ and utilizes $\mathbf{Q}$, $\mathbf{K}$ to calculate the attention matrix $\mathbf{A}^e$, i.e., $\mathbf{A}^e = softmax(\frac{\mathbf{Q}\mathbf{K}^\top}{\sqrt{d_1}})$, where $d_1$ is the dimension of $\mathbf{Q}$, $\mathbf{K}$. However, this traditional approach results in more trainable parameters. Given that the PTM inherently possesses strong representational capabilities, and the attention matrix $\mathbf{A}^o$ in the PTM encapsulates the relationships among features. We propose that the $\mathbf{A}^e$ can be replaced by the one in PTM (i.e., $\mathbf{A}^e = \mathbf{A}^o$) without losing the plasticity. Finally, $\boldsymbol{f}_t^{\hat{e}_l}$ is treated as $\mathbf{V}$, and the output is computed as $\boldsymbol{f}_t^{\bar{e}_l} = \mathbf{A}_t^{o_l} \boldsymbol{f}_t^{\hat{e}_l}$. This attention module functions as a unique form of cross-attention between the trained network and the new sub-network.

The learnable transfer gate addresses the issue of non-smooth representation shift by designing a transfer gate to transfer the features from the trained model to the sub-network. A naive approach would be to directly sum the old and new features. However, we have found that shallow and deep layers in the trained model exhibit different characteristics, and the sub-network should selectively inherit the knowledge from the trained model. Therefore, we have developed a learnable transfer gate for each block to preserve essential knowledge and enhance plasticity. Specifically, the gate includes downsample and upsample layers identical to those in the sub-network, followed by a sigmoid activation function, which constrains its output to a range between 0 and 1. The input of the $l$-th gate for task $t$, denoted as $\boldsymbol{f}_{t-1}^{o_l}$, is the output of the $l$-th block of the trained network, and the output is the weight mask $\mathbf{M}_t^l$. Finally, we fuse the features of the trained network and the sub-network as follows: $\boldsymbol{f}_t^{e_l} = (1 - \mathbf{M}_t^l)\boldsymbol{f}_t^{\bar{e}_l} + \mathbf{M}_t^l \boldsymbol{f}_{t-1}^{o_l}$.

Generally, the sub-network and the transfer gate can be integrated into each block. However, we have found that fusing the features of the last block reduces plasticity. Therefore, we have removed the transfer gate from the $L$-th block and independently introduced two lightweight linear layers in place of the original Feedforward Network (FFN) in the $L$-th block to enhance plasticity.

During inference, we obtain the embedding representation for the $t$-th task as $\mathbf{F}_t = [\boldsymbol{f}_0^{e_L}, \boldsymbol{f}_1^{e_L}, ..., \boldsymbol{f}_t^{e_L}]$. Following the EASE Zhou et al. (2024c), we employ the "semantic-guided prototype complement strategy" to synthesize new features for old classes without accessing any old class instance and classify them using Formula 4. More details can be found in the supplementary A.1.

### 3.3 MARGIN-CE LOSS

**Inconsistency between stage-wise sub-problem optimization and global inference.** We train our *IPA* model by cross-entropy loss in a stage-wise manner, in each stage the model being optimized individually towards the involved classes in the current stage. A potential problem of such training manner is the inconsistent separation granularity between training and inference phases. More fine-grained classification between all involved categories during inference demands more discriminative representation learning than that in one training stage with a small portion of categories. To alleviate such problem, we propose the Margin-CE loss to optimize the representation learning and enhance the separability between classes, which is inspired by SVM classifier Platt (1998).

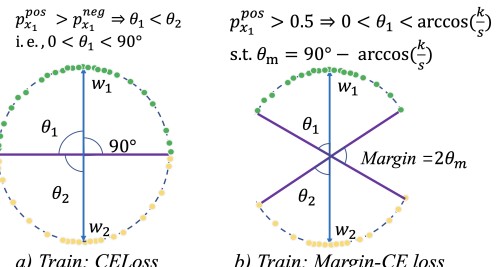

a) Train: CELoss    b) Train: Margin-CE loss

Figure 4: Compare the differences between CELoss and Margin-CE loss during training.

**Margin-CE loss.** Similar to SVM, our proposed Margin-CE loss imposes a margin between the classification boundaries for different classes to optimize the inter-class separability, thereby yielding more discriminative representation learning (see Figure 4). Specifically, our Margin-CE loss

introduces a logit anchor $k$ to cross-entropy loss and defines the losses for the positive class and negative classes respectively:

$$L^{pos} = -\sum_i y_i \log(p_i^{pos}), \quad L^{neg} = -\log(p^{neg}) \tag{5}$$

Where, $i$ is the groundtruth label, and the $p_i^{pos}, p^{neg}$ calculate as follows:

$$p_i^{pos} = \frac{e^{z_i}}{e^{z_i} + e^k} = \frac{e^{s \cdot \cos(\mathbf{w}_i, \Phi(\mathbf{x}))}}{e^{s \cdot \cos(\mathbf{w}_i, \Phi(\mathbf{x}))} + e^k} \tag{6}$$

$$p^{neg} = \frac{e^{-k}}{\sum_{j,j \neq i}^C e^{z_j} + e^{-k}} = \frac{e^{-k}}{\sum_{j,j \neq i} e^{s \cdot \cos(\mathbf{w}_j, \Phi(\mathbf{x}))} + e^{-k}} \tag{7}$$

Additionally, most CIL tasks involve single-label classification. Therefore, we can simplify Equation 5 for single-label classification tasks as follows:

$$L^{pos} = -\log(p^{pos}), \quad L^{neg} = -\log(p^{neg}) \tag{8}$$

Figure 4 illustrates the differences between CELoss and Margin-CE loss. Here, we consider binary classification as an example and let $\mathbf{w}_1, \mathbf{w}_2 \in \mathbb{R}^{2 \times 2}$ represent the classifier weights for the first and second classes, respectively. For the image $\mathbf{x}_1$, which belong to the first class, let $\mathbf{f}_{x_1} = \Phi(\mathbf{x}_1)$. From Formula 3, it can be inferred that as long as $\mathbf{f}_{x_1}$ falls within the upper half of the region in Figure 4.a, the class will be predicted correctly. In contrast, our Margin-CE Loss requires that $\mathbf{f}_{x_1}$ fall within the upper half of the region in Figure 4.b to be classified as a correct prediction. The difference between these two conditions introduces a margin, which provides stronger supervision during training and results in more discriminative features. Based on Formula 6 and Formula 7, we note that the logit anchor must satisfy $0 \leq k < s$ because $\cos(\mathbf{w}_i, \Phi(\mathbf{x})) \in [-1, 1]$.

In order to balance $L^{pos}$ and $L^{neg}$, we set $\lambda$ as the loss weight, and the Margin-CE loss is defined as:

$$L_m = L^{pos} + \lambda L^{neg} \tag{9}$$

Given that the pre-trained model has a good feature distribution, we can alleviate overfitting by transferring it to the current stage. Therefore, knowledge distillation (KD) is utilized, resulting in the final loss function:

$$L_{final} = L_m + \alpha L_{kd} \tag{10}$$

Here, $\alpha$ is the loss weight of $L_{kd}$ with $L_{kd}$ being the loss of the final embedding (such as the final [CLS] token in ViT) between the PTM and the sub-network. For simplicity, we use cosine distance as the metric for $L_{kd}$.

## 4 EXPERIMENTS

In this section, to illustrate its superiority, we compare *DRL* with state-of-the-art methods on six benchmark datasets across different pre-trained models and data split settings. Moreover, an ablation study is conducted, which demonstrates the robustness of our proposed approach. Finally, the paper also provides visualization and parameter analysis, illustrating the effectiveness of *DRL*. Additional experimental results are included in the supplementary material (see Section A.3).

**Datasets.** We evaluate the performance on six datasets, such as CIFAR100 Krizhevsky et al. (2009), ImageNet-R Hendrycks et al. (2021a), and ImageNet-A Hendrycks et al. (2021b), ObjectNet Barbu et al. (2019), OmniBench Zhang et al. (2022), and VTAB Zhai et al. (2019). These datasets include typical CIL benchmarks (the first two datasets) as well as out-of-distribution datasets (the last four datasets) which have a large domain gap with ImageNet (i.e., the pre-trained model's dataset). There are 100 classes in CIFAR100, 200 classes in ImageNet-R, ImageNet-A, ObjectNet, 300 classes in OmniBench, and 50 classes in VTAB. Ablations and visualizations are primarily conducted on ImageNet-A and VTAB because ImageNet-A contains challenging samples that ImageNet pre-trained models cannot handle, while VTAB contains diverse classes from multiple complex realms. In accordance with the benchmark settings in Rebuffi et al. (2017); Wang et al. (2022d); Zhou et al. (2023a), the class split is denoted by 'B-$m$ Inc-$n$'. Here, $m$ is the number of classes in the initial stage, and $n$ is the number of classes in each incremental stage.

Table 1: Comparison of average and last Top-1 accuracy across six benchmark datasets using ViT-B/16-IN21K as the pre-trained model. 'IN-R/A' stands for 'ImageNet-R/A'. Best performances are highlighted in bold. All methods are implemented without using exemplars.

| Method | CIFAR B0 Inc5 | | IN-R B0 Inc5 | | IN-A B0 Inc20 | | ObjNet B0 Inc10 | | Omni B0 Inc30 | | VTAB B0 Inc10 | |
|---|---|---|---|---|---|---|---|---|---|---|---|---|
| | $\bar{\mathcal{A}}$ | $\mathcal{A}_T$ | $\bar{\mathcal{A}}$ | $\mathcal{A}_T$ | $\bar{\mathcal{A}}$ | $\mathcal{A}_T$ | $\bar{\mathcal{A}}$ | $\mathcal{A}_T$ | $\bar{\mathcal{A}}$ | $\mathcal{A}_T$ | $\bar{\mathcal{A}}$ | $\mathcal{A}_T$ |
| Finetune | 38.90 | 20.17 | 21.61 | 10.79 | 24.28 | 14.51 | 19.14 | 8.73 | 23.61 | 10.57 | 34.95 | 21.25 |
| Finetune Adapter Chen et al. (2022) | 60.51 | 49.32 | 47.59 | 40.28 | 45.41 | 41.10 | 50.22 | 35.95 | 62.32 | 50.53 | 48.91 | 45.12 |
| LwF Li & Hoiem (2017) | 46.29 | 41.07 | 39.93 | 26.47 | 37.75 | 26.84 | 33.01 | 20.65 | 47.14 | 33.95 | 40.48 | 27.54 |
| SDC Yu et al. (2020) | 68.21 | 63.05 | 52.17 | 49.20 | 29.11 | 26.63 | 39.04 | 29.06 | 60.94 | 50.28 | 45.06 | 22.50 |
| L2P Wang et al. (2022d) | 85.94 | 79.93 | 66.53 | 59.22 | 49.39 | 41.71 | 63.78 | 52.19 | 73.36 | 64.69 | 77.11 | 77.10 |
| DualPrompt Wang et al. (2022c) | 87.87 | 81.15 | 63.31 | 55.22 | 53.71 | 41.67 | 59.27 | 49.33 | 73.92 | 65.52 | 83.36 | 81.23 |
| CODA-Prompt Smith et al. (2023) | 89.11 | 81.96 | 64.42 | 55.08 | 53.54 | 42.73 | 66.07 | 53.29 | 77.03 | 68.09 | 83.90 | 83.02 |
| SimpleCIL Zhou et al. (2024a) | 87.57 | 81.26 | 62.58 | 54.55 | 59.77 | 48.91 | 65.45 | 53.59 | 79.34 | 73.15 | 85.99 | 84.38 |
| APER w/ Finetune Zhou et al. (2024a) | 87.67 | 81.27 | 70.51 | 62.42 | 61.01 | 49.57 | 61.41 | 48.34 | 73.02 | 65.03 | 87.47 | 80.44 |
| APER w/ VPT-S Zhou et al. (2024a) | 90.43 | 84.57 | 66.63 | 58.32 | 58.39 | 47.20 | 64.54 | 52.53 | 79.63 | 73.68 | 87.15 | 85.36 |
| APER w/ Adapter [paper] Zhou et al. (2024a) | 90.65 | 85.15 | 72.35 | 64.33 | 60.47 | 49.37 | 67.18 | 55.24 | 80.75 | 74.37 | 85.95 | 84.35 |
| APER w/ Adapter [code] Zhou et al. (2024a) | 91.20 | 85.41 | 70.91 | 62.28 | 64.63 | 53.85 | 69.86 | 57.22 | 80.89 | 74.45 | 90.20 | 86.16 |
| EASE Zhou et al. (2024c) | 91.51 | 85.80 | 78.31 | 70.58 | 65.34 | 55.04 | 70.84 | 57.86 | 81.11 | 74.85 | 93.61 | 93.55 |
| DRL | **92.01** | **86.91** | **78.87** | **72.20** | **68.79** | **59.25** | **72.69** | **60.29** | **81.26** | **74.98** | **95.73** | **95.01** |

**Evaluation Metric.** Following the benchmark protocol Rebuffi et al. (2017), we denote the Top-1 accuracy after the $t$-th stage as $\mathcal{A}_t$. We use $\mathcal{A}_T$ (the performance after the last stage) and $\bar{\mathcal{A}} = \frac{1}{T}\Sigma_{t=1}^T \mathcal{A}_t$ (average performance along incremental stages) as measurements.

**Comparison methods.** For comparison, we select state-of-the-art PTM-based CIL methods: L2P Wang et al. (2022d), DualPrompt Wang et al. (2022c), CODA-Prompt Smith et al. (2023), APER Zhou et al. (2024a), and EASE Zhou et al. (2024c). Our method is also compared to conventional CIL methods, all utilizing the same PTM, such as LwF Li & Hoiem (2017), SDC Yu et al. (2020), iCaRL Rebuffi et al. (2017), DER Yan et al. (2021), FOSTER Wang et al. (2022a), and MEMO Zhou et al. (2023b). It is important to note that all the methods are initialized with a same PTM.

**Training details.** Experiments are conducted on an NVIDIA V100 GPU, and other methods are reproduced using PyTorch Paszke et al. (2019). Following Wang et al. (2022d); Zhou et al. (2024a), two representative models, ViT-B/16-IN21K and ViT-B/16-IN1K, are considered as the pre-trained models. These models are pre-trained on ImageNet21K and ImageNet1K, respectively. For *DRL*, the model is trained using an SGD Robbins & Monro (1951) optimizer with a batch size of 48 over 20 epochs. A learning rate of 0.01 is employed with cosine annealing, while $\alpha$ and $\lambda$ are set to 0.5 and 2, respectively. More details are included in the supplementary material A.2.

### 4.1 COMPARISON TO OTHER METHODS

This section presents a comprehensive comparison of *DRL* with other state-of-the-art methods using ViT-B/16-IN21K on six benchmark datasets. As illustrated in Table 1, *DRL* consistently outperforms all other methods across the benchmarks. Notably, *DRL* significantly exceeds the performance of existing state-of-the-art methods such as EASE, APER, and DualPrompt. On out-of-distribution datasets with a large domain gap from ImageNet, *DRL* shows an approximate 2% improvement over the current SOTA, EASE. For instance, on ImageNet-A, VTab, and ObjectNet, *DRL* achieves $\bar{\mathcal{A}}$ scores of 68.79%, 95.73%, and 72.69%, outperforming the current SOTA by 3.45%, 2.12%, and 1.85%, respectively. In terms of $\mathcal{A}_T$, *DRL* records scores of 59.25%, 95.01%, and 60.29%, surpassing the current SOTA by 4.21%, 1.46%, and 2.43%, respectively.

Additionally, we also include performance results using ViT-B/16-IN1K in Table 2. *DRL* notably outperforms the second-best method by 2.37% on ObjNet and 3.68% on ImageNet-A. The results in Tables 1 and 2 demonstrate that *DRL* consistently outperforms the current SOTA across different data splits and pre-trained models.

### 4.2 ABLUTION STUDY

In this section, we conduct an ablation study to investigate the effectiveness of each component in *DRL*.

We display the effectiveness of different components in Table 3. Here, we take EASE as our baseline, and the 'CE, KD, BCE, MCE' represent the model trained with '$L_{ce}$, $L_{kd}$, binary cross-entropy loss,

$L_m$', respectively. *DRL* stands for 'IPA+MCE+KD'. To ensure a fair comparison, given that *IPA*'s final loss function includes $L_{kd}$, we conducted an additional experiment labeled 'Baseline+CE+KD'. The results indicate that our *IPA* with transfer gate significantly improves performance and yields comparable results to the baseline (refer to 'IPA+CE' and 'Baseline+CE'). Furthermore, the proposed Margin-CE loss proves effective, achieving a 1.45% improvement on ImageNet-A (refer to 'IPA+CE+KD' and 'DRL').

The ablation study of loss weight $\alpha$ and $\lambda$ are showed in Table 4, reflecting the stability of our Margin-CE loss for $\lambda \in [1, 3]$

Table 2: Comparison to SOTA classical CIL methods with ViT-B/16-IN1K as the pre-trained model. All methods are deployed without exemplars.

| Method | ObjNet B0 Inc20 | | IN-A B0 Inc20 | |
|---|---|---|---|---|
| | $\bar{A}$ | $A_T$ | $\bar{A}$ | $A_T$ |
| iCaRL Rebuffi et al. (2017) | 33.43 | 19.18 | 29.22 | 16.16 |
| LUCIR Hou et al. (2019) | 41.17 | 25.89 | 31.09 | 18.59 |
| DER Yan et al. (2021) | 35.47 | 23.19 | 33.85 | 22.27 |
| FOSTER Wang et al. (2022a) | 37.83 | 25.07 | 34.82 | 23.01 |
| MEMO Zhou et al. (2023b) | 38.52 | 25.41 | 36.37 | 24.46 |
| FACT Zhou et al. (2022) | 60.59 | 50.96 | 60.13 | 49.82 |
| SimpleCIL Zhou et al. (2024a) | 62.11 | 51.13 | 59.67 | 49.44 |
| APER w/ SSF Zhou et al. (2024a) | 68.75 | 56.79 | 63.59 | 52.67 |
| EASE Zhou et al. (2024c) | 70.44 | 58.37 | 65.74 | 57.28 |
| DRL | **72.81** | **61.00** | **69.42** | **59.97** |

Table 3: Effectiveness of each component in the proposed approach on Imagenet-A and VTab using ViT-B/16-IN21K as the pre-trained model. All methods are deployed without exemplars.

| | IN-A B0 Inc20 | | VTAB B0 Inc10 | |
|---|---|---|---|---|
| | $\bar{A}$ | $A_T$ | $\bar{A}$ | $A_T$ |
| Baseline+CE | 65.34 | 55.04 | 93.56 | 93.58 |
| Baseline+CE+KD | 44.12 | 33.18 | 86.59 | 85.15 |
| IPA+CE+w/o Gate | 61.58 | 51.09 | 93.24 | 91.68 |
| IPA+CE+KD w/o Gate | 66.45 | 55.62 | 94.31 | 93.30 |
| IPA+CE+KD | 67.24 | 57.12 | 94.72 | 94.03 |
| IPA+CE+KD+BCE | 67.32 | 56.92 | 94.55 | 93.04 |
| DRL(IPA+MCE+KD) | **68.96** | **59.38** | **95.73** | **95.01** |

Table 4: Effectiveness of the loss weight on Imagenet-A and VTAB using ViT-B/16-IN21K as the pre-trained model.

| Method | $\alpha$ | $\lambda$ | ImageNet-A B0 Inc20 | | VTAB B0 Inc20 | |
|---|---|---|---|---|---|---|
| | | | $\bar{A}$ | $A_T$ | $\bar{A}$ | $A_T$ |
| DRL | 0 | 2 | 67.285 | 57.01 | 94.90 | 93.97 |
| DRL | 0.5 | 2 | 68.960 | 59.38 | 95.73 | 95.01 |
| DRL | 1 | 2 | 68.735 | 58.72 | 95.21 | 94.30 |
| DRL | 3 | 2 | 68.112 | 57.47 | 94.73 | 93.72 |
| DRL | 5 | 2 | 67.508 | 56.35 | 94.17 | 93.22 |
| DRL | 0.5 | 0.5 | 65.52 | 54.97 | 94.65 | 93.75 |
| DRL | 0.5 | 1 | 68.46 | 58.07 | 95.00 | 94.16 |
| DRL | 0.5 | 2 | **68.96** | **59.38** | **95.73** | **95.01** |
| DRL | 0.5 | 3 | 68.17 | 58.06 | 95.40 | 94.49 |
| DRL | 0.5 | 5 | 67.16 | 57.47 | 94.93 | 94.16 |

Table 5: Generalization experiments of Margin-CE loss on Imagenet-A and VTAB utilizing ViT-B/16-IN21K as the pre-trained model. We simply replace the CELoss in the original methods with our Margin-CE loss.

| | ImageNet-A B0 Inc20 | | VTAB B0 Inc10 | |
|---|---|---|---|---|
| | $\bar{A}$ | $A_T$ | $\bar{A}$ | $A_T$ |
| APER+CE | 64.63 | 53.85 | 90.20 | 86.16 |
| ESN+CE | 52.66 | 41.54 | 86.34 | 69.23 |
| EASE+CE | 65.34 | 55.04 | 93.56 | 93.55 |
| APER+MCE | 65.54 | 54.25(**+0.4**) | 92.57 | 88.84(**+2.68**) |
| ESN+MCE | 53.94 | 42.92(**+1.38**) | 88.77 | 72.61(**+3.38**) |
| EASE+MCE | 67.71 | 58.85(**+3.81**) | 95.36 | 94.57(**+0.98**) |

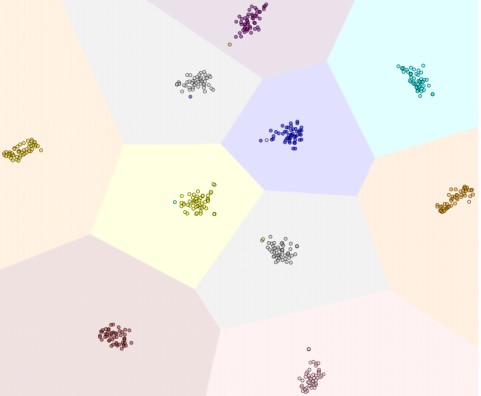

Figure 5: *DRL*: t-SNE Visualization of stage 1 for VTAB Dataset with B0 Inc10 Setting

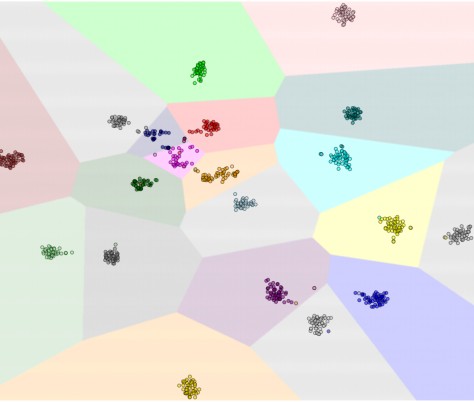

Figure 6: *DRL*: t-SNE Visualization of stage 2 for VTAB Dataset with B0 Inc10 Setting

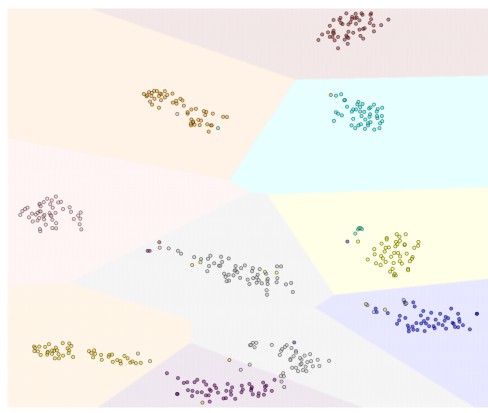 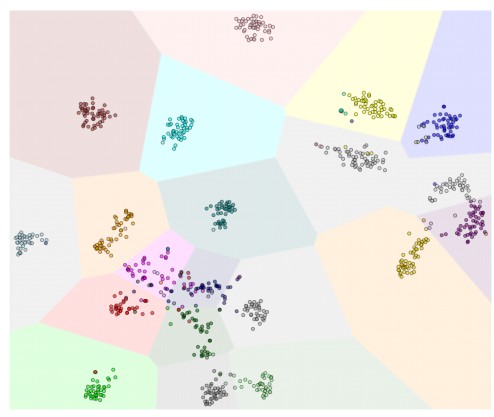

Figure 7: *DRL* w/o Margin-CE loss: t-SNE Visualization of stage 1 for VTAB Dataset with B0 Inc10 Setting

Figure 8: *DRL* w/o Margin-CE loss: t-SNE Visualization of stage 2 for VTAB Dataset with B0 Inc10 Setting

### 4.3 MORE INVESTIGATION OF *DRL*

**Efficient analysis.** This section analyzes the efficiency of our approach by examining the number of network parameters during training and testing. Let '1B' denote the total number of parameters for ViT-B/16. Figure 2 demonstrates that our *IPA* comprises only 0.6% of trainable parameters, while requiring only $(1 + 0.006t)$B parameters for inference, indicating efficiency in both training and testing phases. More results show in Figure 1.

**Visualization.** In this section, we employ t-SNE Van der Maaten & Hinton (2008) to visualize the learned decision boundaries on the VTAB dataset between two incremental stages, as illustrated in Figure 5 and 6. For clarity, we represent the classes from the first and second incremental stages, with each stage comprising 10 classes (VTAB B0 Inc10). As inferred from these figures, *DRL* exhibits competitive performance, effectively separating instances into their respective classes. Furthermore, Figure 7 and Figure 8 indicate that the representation is less discriminal without Margin-CE loss ('DRL w/o Margin-CE loss' refers to training with CELoss instead of Margin-CE loss).

**Generalization experiments.** To verify Margin-CE loss's generalization, we integrate it into various methods. We selected three representative methods: APER, ESN, and EASE. APER is a prototype-based classifier similar to ours, ESN is network-based, and EASE represents the current state-of-the-art. All employed Cross-Entropy Loss (CELoss) for training. Experiments were conducted by replacing the original methods' CELoss with Margin-CE loss. Table 5 shows that Margin-CE loss consistently achieves significant improvement.

## 5 CONCLUSION

In this paper, we propose a novel non-rehearsal CIL method, Discriminative Representation Learning (*DRL*), which consists of an *IPA* and a Margin-CE loss. *IPA* chieves a better stability-plasticity trade-off with high efficiency. Experiments on various datasets demonstrate that our method achieves new state-of-the-art performance. Overall, our work presents a promising direction for future research in CIL and its application in real-world scenarios.

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

# A   SUPPLEMENTARY MATERIALS

## A.1   INFERENCE DETAILS

Facing the continual data stream, we freeze the trained model $\Phi(\cdot)$ and extract the center $\mathbf{c}$ of each class:

$$\mathbf{c}_i = \frac{1}{N} \sum_j \mathbb{I}(y_j = i)\Phi(\mathbf{x}_j) \tag{A1}$$

Here, $N$ is the number of images in class $i$, and $\mathbb{I}(\cdot)$ is the indicator function that outputs 1 if the expression holds and 0 otherwise. The embedding representation for class $i$ in the $t$-th task is denoted as $\mathbf{F}_t^i = [\boldsymbol{f}_0^{e_L^i}, \boldsymbol{f}_1^{e_L^i}, ..., \boldsymbol{f}_t^{e_L^i}]$, where $\boldsymbol{f}_t^{e_L^i}$ is the embedded [CLS] token in the $L$-th block. Note that the for the classes in the $t$-1 stage, we cannot obtain the $\boldsymbol{f}_t^{e_L^i}$ since we do not have access to the previous data. Therefor, we employ the "semantic-guided prototype complement strategy Zhou et al. (2024c)" to synthesize new features for old classes without accessing any instances of those classes.

## A.2   TRAINING DETAILS

For *DRL*, the model is trained using an SGD optimizer,with momentum and weight decay parameters set to 0.9 and 0.0005, respectively. For all six benchmarks, $k$ is set to 2, and $r$ is set to 48 in $\mathbf{W}_{down}$ and $\mathbf{W}_{up}$. In the $L$-th block of the sub-network, the first lightweight linear layer is $\mathbf{W}_{first} \in \mathbb{R}^{768 \times 768}$ and the second linear layer is $\mathbf{W}_{second} \in \mathbb{R}^{768 \times 768}$.

## A.3   EXTRA EXPERIMENTS

In this section, we conduct extra experiments to verify the effectiveness of our method.

There are many methods to fuse old and new features using the transfer gate (Section 3.2). We consider three approaches and investigate the effectiveness of our learning transfer gate, presenting the results in Table A1. Here, 'DRL+sum' denotes $\boldsymbol{f}_t^{e_l} = \boldsymbol{f}_t^{\bar{e}_l} + \boldsymbol{f}_{t-1}^{o_l}$, 'DRL + mask-PTM' denotes $\boldsymbol{f}_t^{e_l} = \boldsymbol{f}_t^{\bar{e}_l} + \mathbf{M}_t^l \boldsymbol{f}_{t-1}^{o_l}$, and 'DRL + mask-ALL' denotes $\boldsymbol{f}_t^{e_l} = (1 - \mathbf{M}_t^l)\boldsymbol{f}_t^{\bar{e}_l} + \mathbf{M}_t^l \boldsymbol{f}_{t-1}^{o_l}$. The results confirm that our learning transfer gate is effective, achieving a performance increase of 1.04% compared to directly summing the old features (i.e., 'DRL + mask-ALL' vs. 'DRL + sum').

Secondly, Table A2 indicates that utilizing $\mathbf{A}^o$ to replace $\mathbf{A}^e$ does not affect the plasticity. The attention matrix $\mathbf{A}^o$ in PTM relationships among the features that can be directly reused to our sub-network. Here, 'reuse attention' denotes $\mathbf{A}^e = \mathbf{A}^o$, 'self-attention' denotes $\mathbf{A}^e = softmax(\frac{\mathbf{f}_t^{\bar{e}} \mathbf{f}_t^{\bar{e}\top}}{\sqrt{d_1}})$, 'project attention' denotes the standard attention with learned $\mathbf{Q}, \mathbf{K}, \mathbf{V}$. Compared to the two methods 'self-attention' and 'project attention', using $\mathbf{A}^0$ to replace $\mathbf{A}^e$ can further reduce the number of training parameters and the computational complexity of the network, thereby making our *IPA* more efficient.

Table A1: Ablation experiments on the gate branch on Imagenet-A and VTAB using ViT-B/16-IN21K as the pre-trained model.

| | ImageNet-A B0 Inc20 | | VTAB B0 Inc10 | |
| --- | --- | --- | --- | --- |
| | $\bar{\mathcal{A}}$ | $\mathcal{A}_T$ | $\bar{\mathcal{A}}$ | $\mathcal{A}_T$ |
| DRL + sum | 68.349 | 58.52 | 94.50 | 93.97 |
| DRL + mask-PTM | 67.296 | 57.27 | 94.52 | 93.83 |
| DRL + mask-ALL | 68.960 | 59.38 (+0.86) | 95.73 | 95.01 (+1.04) |

Table A2: Ablation experiments on the attention strategy in adapter on ImageNet-A and VTAB using ViT-B/16-IN21K as the pre-trained model.

| | ImageNet-A B0 Inc20 | | VTAB B0 Inc10 | |
| --- | --- | --- | --- | --- |
| | $\bar{\mathcal{A}}$ | $\mathcal{A}_T$ | $\bar{\mathcal{A}}$ | $\mathcal{A}_T$ |
| project+attention | 68.78 | 59.76 | 95.57 | 95.14 |
| self-attention | 68.62 | 59.56 | 95.74 | 95.05 |
| reuse attention | 68.96 | 59.38 | 95.73 | 95.01 |

Furthermore, we investigate the influence of the logit anchor $k$ with different values. Noting that the anchor must satisfy $0 \leq k < s0$, where $s$ is a learning scale factor, we conduct experiments with values in the set $\{0, 0.5, 1, 2, 3, 5, 10, 20\}$. Table A3 shows that performance remains stable when the anchor is in the range [0,5]. Based on Formula 6, If the value of $k$ is too close to $s$, the experimental results will not be favorable. The experimental results also confirm this conclusion.

Table A3: Ablation experiments on the anchor $k$ using ViT-B/16-IN21K as the pre-trained model.

| Method | $k$ | learned s | ImageNet-A B0 Inc20 | | learned s | VTAB B0 Inc10 | |
|---|---|---|---|---|---|---|---|
| | | | $\bar{\mathcal{A}}$ | $\mathcal{A}_T$ | | $\bar{\mathcal{A}}$ | $\mathcal{A}_T$ |
| DRL | 0 | 12.62 | 68.07 | 58.27 | 9.25 | 95.72 | 94.78 |
| DRL | 0.5 | 13.13 | 68.97 | 58.4 | 9.88 | 95.82 | 94.83 |
| DRL | 1 | 13.77 | 68.98 | 59.26 | 9.48 | 95.76 | 94.95 |
| DRL | 2 | 14.41 | 68.96 | 59.38 | 11.09 | 95.73 | 95.01 |
| DRL | 3 | 14.59 | 69.41 | 59.59 | 12.25 | 95.54 | 94.74 |
| DRL | 5 | 15.62 | 68.88 | 58.54 | 12.41 | 95.55 | 94.79 |
| DRL | 10 | 19.77 | 67.36 | 57.28 | 16.56 | 95.59 | 94.96 |
| DRL | 20 | 23.04 | 66.20 | 55.51 | 23.32 | 95.49 | 94.83 |

Table A4: Ablation experiments on the variations in the feature dimension of the last block using ViT-B/16-IN21K as the pre-trained model.

| Method | ImageNet-A B0 Inc20 | | VTAB B0 Inc10 | |
|---|---|---|---|---|
| | $\bar{\mathcal{A}}$ | $\mathcal{A}_T$ | $\bar{\mathcal{A}}$ | $\mathcal{A}_T$ |
| 768→192→768 | 68.91 | 59.23 | 95.616 | 94.88 |
| 768→384→768 | 68.78 | 59.04 | 95.506 | 94.92 |
| 768→768→768 | 68.96 | 59.38 | 95.73 | 95.01 |
| 768→1536→768 | 68.98 | 59.24 | 95.632 | 95.14 |
| 768→2304→768 | 68.98 | 59.83 | 95.560 | 94.94 |

Finally, Table A4 presents the results of experiments conducted with different configurations of the two lightweight linear layers used to replace the feedforward network (FFN) in the $L$-th block. Here, '768→384→768' denotes the first linear layer is $\mathbf{W}_{first} \in \mathbb{R}^{768 \times 384}$ and the second linear layer is $\mathbf{W}_{second} \in \mathbb{R}^{384 \times 768}$, and so on for others. The results reveal that utilizing '768→768→768' can perform well in our *IPA*. This also demonstrates the effectiveness of our *DRL*, as the learned representation is more discriminative and achieves good plasticity with fewer training parameters.

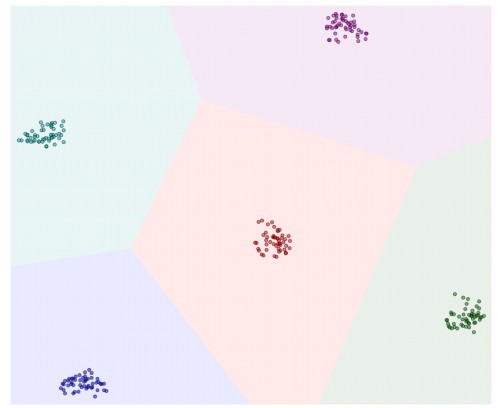

Figure A1: *DRL*: t-SNE Visualization of stage 1 for CIFAR100 Dataset with B0 Inc5 Setting

Figure A2: *DRL*: t-SNE Visualization of stage 2 for CIFAR100 Dataset with B0 Inc5 Setting

## A.4 VISUALIZATION.

In this section, we also employ t-SNE Van der Maaten & Hinton (2008) to visualize the learned decision boundaries on the CIFAR100 dataset between two incremental stages, as illustrated in Figure A1 and A2. Each stage comprises 5 classes (CIFAR100 B0 Inc5). Based on these figures, *DRL* demonstrates competitive performance by effectively distinguishing instances into their respective classes.

