# OpenReview forum: "DRL: DISCRIMINATIVE REPRESENTATION LEARNING FOR CLASS INCREMENTAL LEARNING"
_ICLR.cc/2025/Conference — ICLR 2025 Conference Withdrawn Submission_

### Official Review · Reviewer_Uba7 · 2024-10-28

**Soundness:** 2
**Presentation:** 3
**Contribution:** 2
**Rating:** 5
**Confidence:** 4

**Summary:**

This paper introduces an approach to non-rehearsal class incremental learning (CIL) through a method called Discriminative Representation Learning (DRL). The core of DRL consists of an Incremental Parallel Adapter (IPA) and a Margin Cross-Entropy (Margin-CE) loss. The IPA is designed to enhance CIL by allowing the model to incrementally learn new classes without forgetting prior knowledge, using a pre-trained model to minimize parameter increase. The Margin-CE loss is introduced to enforce distinct classification boundaries, which improves inter-class separability and addresses inconsistencies between stage-wise optimization and global inference.

**Strengths:**

1. The IPA approach, which utilizes a PTM and dynamically learns adapters for each incremental stage, is resource-efficient. Additionally, using a pre-trained model with a minimal parameter increase (0.6% per stage) underscores DRL’s efficiency.
2. The Margin-CE loss enhances class separability, optimizing the model’s discriminative power and addressing issues related to inconsistent stage-wise optimization.
3. Extensive experiments demonstrate DRL’s good performance.

**Weaknesses:**

1. While the authors claim that "Margin-CE loss achieves a better stability-plasticity trade-off with high efficiency" in Line 84, there is no evidence in the paper supporting this improved stability-plasticity trade-off.
2. In Lines 86-87, the authors make assertions without providing supporting arguments, which may leave readers confused and doubtful about the claim's validity.
3. The paper lacks comparisons with some baselines, such as SLCA [1], RanPAC [2], and LAE [3].
4. Although the paper mentions efficiency, it does not provide a detailed comparative analysis of DRL’s computational or memory requirements relative to other CIL methods. A more explicit evaluation of resource consumption would be valuable, especially for practitioners with hardware constraints.

[1] SLCA: Slow Learner with Classifier Alignment for Continual Learning on a Pre-trained Model. ICCV 2023.

[2] RanPAC: Random Projections and Pre-trained Models for Continual Learning. NeurIPS 2023.

[3] A Unified Continual Learning Framework with General Parameter-Efficient Tuning. ICCV 2023.

**Questions:**

See weaknesses.

---

### Official Review · Reviewer_3c56 · 2024-10-28

**Soundness:** 2
**Presentation:** 1
**Contribution:** 2
**Rating:** 5
**Confidence:** 3

**Summary:**

This paper focuses on utilizing large pre-trained models (PTM) for class incremental learning without access to previously learned datasets. By making use of PTM, the model have improved representation learning capability. The model is expanded with an learnable lightweight adaptor at each incremental step to accommodate new knowledge and freezes all weights from previous time step to avoid forgetting. The authors also address the issue of inconsistency between stage-wise sub-problem optimization and global inference by proposing a margin cross-entropy loss, which imposes a margin between class boundaries to learn more discriminative representations and improve separability. Experimental results demonstrate the effectiveness of the proposed approach on six benchmark datasets.

**Strengths:**

The experiments show a strong performance of the proposed model without access to previously learned datasets while not introducing a large amount of new parameters at each new incremental step, thereby is promising in improving the efficiency and applicability of incremental learning in real-world applications.

**Weaknesses:**

* Quality of writing can be further improved:
  * The missing brackets around the citations make it difficult to read
  * Use of the term IPA is confusing, especially in Section 3.2. Sometimes it refers to the whole model, but sometimes the newly inserted block. Does IPA refer to the whole adaptor block added at each incremental step or just the 1x1 convolution layers within the new block?
* Model details: In line 294-297, the authors mentioned the transfer gate from the L-th block is removed and two linear layers are used in place of the FFN instead. However, there is no FFN shown within the adaptors in Figure 3. Does this mean that the L-th block does not have the same structure as the adaptor?
* Effectiveness of reusing attention matrix: Given that reusing $\bm{A}_o$ as $\bm{A}_e$ is part of the proposed model architecture, it would be expected that there is some study of its effectiveness as compared to other approaches in the experiments section of the main paper.
* Comparison with other margin based loss functions: The authors only compared the proposed Margin-CE loss against cross-entropy loss. How about other margin-based loss functions such as hinge loss and large-margin softmax loss [1]?
* Comparative results: The empirical results under Section 4.1 seems incomplete. Specifically, why compare against a different set of baseline methods and only a subset of datasets when using ViT-B/16-IN1K as pre-trained model, as compared to ViT-B/16-IN21K?
* Ablation results: The ablation study seems incomplete. For example, the authors mentioned the effectiveness of IPA with transfer gate by comparing IPA+CE and Baseline+CE but results of IPA+CE is not found. Also, the separate effectiveness of adaptor and gate themselves doesn't seem to be reflected in the results?

[1] W. Liu, Y. Wen, Z. Yu, and M. Yang. Large-margin softmax loss for convolutional neural networks. In ICML, pages 507–516, 2016.

**Questions:**

Please refer to questions in section 'Weaknesses'

---

### Official Review · Reviewer_HQ9r · 2024-11-02

**Soundness:** 2
**Presentation:** 2
**Contribution:** 2
**Rating:** 3
**Confidence:** 4

**Summary:**

The paper proposes a method called Discriminative Representation Learning (DRL) for class incremental learning (CIL). The method addresses the challenges of increasingly large model complexity, non-smooth representation shift, and inconsistency between stage-wise sub-problem optimization and global inference in CIL. DRL is built upon a pre-trained large model and incrementally augments it with a lightweight adapter in each stage of incremental learning. The adapter is responsible for adapting the model to new classes while inheriting the representation capability from the current model. The paper also introduces the Margin-CE loss to enhance the discriminative representation learning and alleviate the issue of training-inference inconsistency.

**Strengths:**

1) The paper proposes an innovative method, DRL, for class incremental learning. The method addresses the challenges of large model complexity, non-smooth representation shift, and inconsistency in CIL.
2) DRL is built upon a pre-trained large model, which allows for efficient and effective incremental learning.
3) The paper introduces the Margin-CE loss, which enhances the discriminative representation learning and improves the separation between classes.

**Weaknesses:**

1. The motivation of this manuscript is not clear. The authors should clearly claim the challenging issues in previous methods.
2. The authors complement the theoretical explanation of the success of the proposed approach.
3. While the Margin-CE adopted in the manuscript seems plausible, it is not exciting.

**Questions:**

1) Could you provide more details on the implementation of DRL, including the architecture of the adapter and the training procedure?
2) Could you provide comparisons and discussions with widely-known CIL baselines in the field to demonstrate the superiority of DRL?

---

### Official Review · Reviewer_j1LB · 2024-11-03

**Soundness:** 3
**Presentation:** 3
**Contribution:** 3
**Rating:** 5
**Confidence:** 4

**Summary:**

This paper introduces a new and efficient way of utilising pre-trained model for class incremental learning. Especially, through Incremental Parallel Adapter (IPA) network, the authors presents a efficient (in terms of memory and inference cost) way to adapt to new tasks and building connection between them on top of pre-trained model. Also the authors introduce simple yet effective CE loss called, Margin-CE, specifically tailored for CIL. The authors extensively validated the proposed algorithm under various benchmarks and various settings, which shows the superiority of their method.

**Strengths:**

1. The paper is well organised and easy to follow.
2. The motivation of each component is compelling. Also each component achieves what it aims at quite well.
3. The literature search is well done. The authors tried to explore and compare the very-recent methods as well which makes their work reliable.

**Weaknesses:**

1. While the most of the paper are compelling and shows great result, I think the connection between this paper and 'Non-rehearsal' CIL is quite weak. I think the proposed method can be used for any kind of CIL, including Rehearsal CIL as well. Is there any specific components especially tailored for 'Non-rehearsal' environment?

2. While both IPA and Margin-CE are compelling, it seems a large portion of improvement comes from Margin-CE. Can you provide the results for EASE/APER+CE+KD or other configurations so that I can clearly see the effectiveness of 'IPA' itself without MCE?

3. This might be general question, but I wonder whether the effectiveness of these methods influenced by the similarity between dataset which PTM trained on and dataset the CIL will be done. Especially, this kind of influence might be stronger in IPA, compared to other methods, since IPA uses un-modified attention map. Can you give an explanation or justification on this concern?

4. While Margin-CE improves the inter-class separability, I think it might degrade the representation ability since it can hurt the intra-class (instance-wise) separability.

I am willing to raise my score if the concerns are resolved.

**Questions:**

See the weakness

---

### Note · Authors · 2024-11-13

I have read and agree with the venue's withdrawal policy on behalf of myself and my co-authors.